**Data Availability Statement:** All relevant data are within the paper and its Supporting Information files.

**Funding:** This work was funded by a grant from the company that provided the tracks for the work flow

# Effects of music advertised to support focus on mood and processing speed

**Joan Orpella**[1,2], **Daniel Liu Bowling**[3,4], **Concetta Tomaino**[5,6], **Pablo Ripollés**[2,7,8]*

**1** Department of Neuroscience, Georgetown University Medical Center, Washington, DC, United States of America, **2** Department of Psychology, New York University, New York, NY, United States of America, **3** Department of Psychiatry & Behavioral Sciences, Stanford University School of Medicine, Stanford, CA, United States of America, **4** Center for Computer Research in Music and Acoustics (CCRMA), Stanford University, Stanford, CA, United States of America, **5** Institute for Music and Neurologic Function, Wartburg, Mount Vernon, NY, United States of America, **6** Lehman College, City University of New York, New York, NY, United States of America, **7** Music and Audio Research Laboratory (MARL), New York University, New York, NY, United States of America, **8** Center for Language, Music, and Emotion (CLaME), New York University, New York, NY, United States of America

⊚ These authors contributed equally to this work.
* pripolles@nyu.edu

## Abstract

While music's effects on emotion are widely appreciated, its effects on cognition are less understood. As mobile devices continue to afford new opportunities to engage with music during work, it is important to understand associated effects on how we feel and perform. Capitalizing on potential benefits, many commercial music platforms advertise content specifically to support attentional focus and concentration. Although already in wide-spread use, the effects of such content remain largely untested. In this online behavioral study, we tested the effects of music advertised to support "work flow" and "deep focus" on mood and performance during a cognitively demanding psychological test (the flanker task). We additionally included a sample of popular hit music representing mainstream musical stimulation and a sample of office noise representing typical background stimulation in a social working environment. Our findings show that, despite similar marketing, only the work flow music gave rise to significant and positively correlated improvements in mood and performance (i.e., faster responses over time, with similar accuracy). Analyses of objective and perceived musical features indicate consistency with the "arousal-mood theory" of music's cognitive impact and provide new insights into how music can be structured to regulate mood and cognition in the general population.

## Introduction

Music is at the core of what it means to be human. It is present in all societies and often reported as one of the most engaging, enjoyable, moving, and satisfying human activities [1–3]. People use music to self-regulate their mood and emotions, engaging with it intentionally to improve wellbeing [4–8]. A large body of research shows that music can increase positive affect [9, 10], reduce stress, mitigate anxiety, and decrease depression [11–14] in clinical (e.g.,

condition to Dr. Ripolles. The funders had no role in study design, data collection and analysis, decision to publish, or preparation of the manuscript.

**Competing interests:** Dr. Tomaino serves as the music therapy advisor at the company that provided the tracks for the work flow condition. Dr. Bowling serves as the neuroscience advisor at the company that provided the tracks for the work flow condition. This work was funded by a grant from the company that provided the tracks for the work flow condition to Dr. Ripollés. Dr. Orpella has no competing interests. This does not alter our adherence to PLOS ONE policy on sharing data and materials.

stroke, dementia, and Parkinson's disease among others; [15, 16]) and healthy populations across the lifespan [7, 17–19]. Neurally, there is compelling evidence that music stimulates brain networks critical to emotion, including core dopaminergic and opioidergic regions within the brain's reward system [20–28] and an extended network of cortical and subcortical structures that likely supports the upregulation of positive mood [22, 29, 30].

Although the positive effects of music on mood and wellbeing are well-established, music's impact on cognition are less clear [31]. There is increasing evidence that music can enhance memory via reward-related mechanisms [32–35], but the effects of music on *attention* are more equivocal, with evidence supporting enhancement, impairment, or no effect across clinical and healthy populations [36–43]. Understanding how music modulates selective attention in particular—the ability to enhance important signals while at the same time suppressing distracting information—is critical, given that people often listen to music *while engaged in other tasks* (e.g., working, studying, exercising, cooking, driving, operating heavy machinery, etc.), a fact that differentiates music listening from wellness practices like exercise or meditation [44]. Anecdotally, this type of "on-task" music listening can be an important source of support during tasks that are perceived as mundane or unpleasant, buffering against emotional and/or psychological stress.

Research examining the effects of background music on selective attention [37–39, 45] has made progress using standard psychological tests, such as the flanker task, which requires participants to selectively attend and respond to the features of a central "target" stimulus while inhibiting responses to nearby "flanking" distractors [46, 47]. For example, in two recent laboratory studies, listening to "joyful" classical music during a flanker task resulted in faster reaction times (RTs) in response to target stimuli flanked by both similar (congruent) and dissimilar (incongruent) distractors [38, 39]. These results suggest that certain types of music can have a general positive effect on processing speed during tasks that require selective attention and distractor conflict resolution.

However, when engaging in real-world tasks that require selective attention listeners do not restrict their listening preferences to classical music. And, while success in modulating mood can be achieved with music of many kinds (e.g., from pop, classical, or rock genres; [4–6], listeners are typically unaware of the ways in which their music choices may impact selective attention, and potentially, overall performance. This is problematic because many musical compositions, especially those created mainly for entertainment or self-expression, arguably aim to capture as much attention as possible. Addressing this problem, many commercial music platforms advertise selections of music for the specific purpose of enhancing attentional focus and concentration while engaged in cognitively demanding tasks. While these tracks are becoming widely used, especially with the success of music streaming platforms, research assessing their effects on cognition is lacking. Here, we address this gap in the literature by testing the effects of music advertised to support focus on mood and performance while participants complete a flanker task.

We focus on two audio conditions: a sample of tracks aimed at enhancing "work flow" offered by a commercially available music therapy app, and a sample of tracks aimed at enhancing "deep focus" offered by a commercially available music streaming platform. These were selected because, despite similar marketing, they exhibit pronounced differences in objective musical features (see "Stimuli" below in Materials and Methods) that can be expected to drive different neural and behavioral responses. For context, we also included a third audio condition comprising popular music to represent mainstream musical stimulation (hypothesized to be distracting), and a fourth audio condition comprising simulated office noise to represent background acoustic stimulation in a typical social working environment (as an ecologically valid baseline for sound during work).

In an online behavioral study (N = 196) we compared the effects of the above audio conditions on mood and performance during a flanker task, as a proxy for straight-forward but cognitively demanding work. In this experimental context, we hypothesized that either work flow or deep focus music would positively modulate mood and performance relative to popular music and office noise. Based on previous literature, we also hypothesized that any positive effects on performance would primarily be reflected in generally increased processing speed [38, 39].

## Materials and methods

### Experimental design

**Participants.** Online participants were randomized to one of four audio conditions ("work flow", "deep focus", "pop hits", and "office noise") during which they completed a six-minute flanker task in the middle of ten minutes of listening. The experiment was coded using PsychoPy [48] and participants were recruited online using Amazon Mechanical Turk. Recruitment was restricted to individuals located in the United States with a record of success completing Mechanical Turk tasks (i.e., at least 100 tasks completed with a 95% approval rate from task requesters). Experiments using participants from Mechanical Turk have been previously demonstrated to replicate results from more traditional participant populations across a variety of cognitive tasks, including flanker tasks [49] and tasks involving complex auditory stimuli [50–53]. A power analysis using MorePower [54] indicated a sample size of at least 44 participants per group for 80% power to find a medium effect size (partial eta$^2$ = 0.06; [37]) in a between-subjects one-way ANOVA with four conditions (the analysis required to assess the effect of audio condition on mood). This statistical test was selected among other possible ones (e.g., effect of audio condition on Flanker accuracy) as it was the one requiring the largest sample size. In addition, previous research using a similar paradigm, assessing different groups as we do here, and finding significant effects has used smaller sample sizes than the one we employed ([37]: two groups, N = 19 and N = 21; [38]: two groups, N = 19, N = 33; [39]: two groups, N = 15, N = 15). We recruited 70 participants per group to account for attrition. This experiment was approved by the Institutional Review Board at New York University. Participants were recruited between March 1$^{st}$ 2022 and May 31$^{st}$ 2022. Written consent was obtained online. To do so, participants clicked on an "accept" button after reading the consent form. Participants were paid for their participation in the study.

**Stimuli.** In this experiment, we aimed to study commercially available music advertised to support attentional focus and concentration. Stimuli were selected to contrast two acoustically distinct types of "focus" music. The first was "work flow" music sampled from a synonymous playlist on a music therapy app [55]. Overall, the work flow music included here was characterized by strong rhythm (e.g., moderately fast tempo with high pulse clarity and low rhythmic complexity), simple tonality (e.g., predominantly major with high key clarity and low melodic and harmonic variation), broadly distributed spectral energy below ~6000 Hz (e.g., centroid, spread, and roll-off), and moderate dynamism (e.g., moderately steep attacks and moderate event density; see Table 1). The second type was "deep focus" music, sampled from a synonymous playlist on a music streaming platform [56]. Overall, this music was relatively minimalistic, being characterized by similarly simple tonality, but weaker rhythm (e.g., slower tempo with lower pulse clarity and higher complexity), lower and more restricted spectral energy (e.g., lower centroid, spread, and roll-off), and more reserved dynamism (e.g., gentler attacks and lower event density; see Table 1). Neither work flow nor deep focus music had lyrics. Further context for evaluating the effects of these distinct types of "focus" music was provided by two additional audio conditions, also implemented to reflect real-world listening.

**Table 1. Musical features of the music stimuli.**

| MUSICAL FEATURE | | MUSIC CONDITION | | | STATISTICS | | | | |
|---|---|---|---|---|---|---|---|---|---|
| Type | Name | Work Flow (WF) | Deep Focus (DF) | Pop Hits (PH) | ANOVAs (DF = 2,33) | | Tukey's HSDs (α = 0.05) | | |
| | | | | | F | p | WF vs. DF | WF vs. PH | DF vs. PH |
| Rhythm | Tempo | 119.250 | 70.250 | 111.438 | 19.365 | 2.73e-6 | ** | | *** |
| | Pulse clarity | 0.545 | 0.219 | 0.483 | 9.073 | 7.00e-4 | * | | * |
| | Fluctuation entropy | 0.963 | 0.985 | 0.979 | 41.480 | 9.87e-10 | *** | *** | * |
| | Fluctuation maximum | 4077.285 | 1010.193 | 2573.015 | 15.899 | 1.46e-5 | *** | . | ** |
| Tonality | Key (% major) | 75.000 | 100.000 | 56.250 | | | | | |
| | Key clarity | 0.689 | 0.866 | 0.693 | 8.051 | 1.40e-3 | . | | * |
| | Mode | 0.117 | 0.19 | 0.045 | 4.598 | 1.73e-2 | | | * |
| | Chromatic complexity | 4.750 | 5.625 | 8.688 | 8.672 | 9.00e-4 | | * | * |
| | HCDF mean | 0.195 | 0.174 | 0.257 | 53.87 | 4.04e-11 | | *** | *** |
| Spectrum | Flux | 33.282 | 15.362 | 33.682 | 90.874 | 3.79e-14 | *** | | *** |
| | Entropy | 0.778 | 0.763 | 0.838 | 111.516 | 2.08e-15 | | *** | *** |
| | Centroid | 2273.033 | 1015.235 | 3431.681 | 175.817 | 2.52e-18 | *** | *** | *** |
| | Spread | 3964.343 | 1712.401 | 4029.667 | 61.767 | 6.99e-12 | *** | | *** |
| | Flatness | 0.130 | 0.038 | 0.247 | 73.788 | 6.62e-13 | * | ** | *** |
| | Roll-off | 5613.889 | 1730.855 | 7783.217 | 139.447 | 8.02e-17 | *** | * | *** |
| | Brightness | 0.278 | 0.177 | 0.525 | 234.807 | 3.06e-20 | *** | * | *** |
| | Zero crossing rate | 492.950 | 418.910 | 1199.087 | 97.762 | 1.36e-14 | | *** | *** |
| Dynamics | Attack time | 0.115 | 0.142 | 0.104 | 3.556 | 3.99e-2 | | | * |
| | Attack slope | 3.174 | 1.908 | 4.018 | 19.756 | 2.28e-6 | * | | *** |
| | Decay time | 0.179 | 0.253 | 0.133 | 2.885 | 7.00e-2 | | | |
| | Decay slope | -1.976 | -1.499 | -3.140 | 22.865 | 5.88e-7 | | * | *** |
| | Event density | 1.931 | 1.085 | 2.348 | 14.186 | 3.58e-5 | . | | * |
| | RMS amplitude | 0.058 | 0.047 | 0.054 | 8.807 | 8.61e-4 | ** | | * |

Values represent means for each condition. HCDF = Harmonic Change Density Function; RMS = Root Mean Square; DF = Degrees of Freedom; HSD = Honest Significant Difference. See Supplementary Material and S1 Table for further details. p<0.1

* = p<0.05

**p<0.001

***p<0.0001 (Tukey corrected).

One was popular hit music, sampled from the "Hot 100" playlist published by an American music magazine [57] in the second week of October 2021. Given prior research on the effects of listening to popular music–especially with lyrics–during work and other complex tasks [58–60], we expected this condition to negatively impact on-task performance. The other condition was simulated "calm office noise", sampled from a synonymous sound generator on a website offering noise stimulation [61]. Our intention with this last audio condition was to represent acoustic stimulation in a naturalistic social working environment. It was preferred to silence because people rarely work in acoustic isolation. Audio files of the work flow tracks are included in supplementary material; the deep focus and pop hits tracks are available commercially.

For each of the three musical audio conditions, we obtained enough music to create four unique stimuli, each comprising at least ten minutes of stimulation. For the work flow condition, our sample comprised four tracks, each just over ten minutes in duration and each composed by a different artist. Each participant in this condition heard only one of the tracks, with

the specific track counterbalanced across participants. For the deep focus condition, our sample comprised sixteen tracks, each approximately three minutes in duration and each composed by a different artist. These were sorted into four sets of four tracks each (satisfying the required minimum of ten minutes; note that even though these sets were approximately twelve minutes long, playback was terminated at 10 minutes; see "Procedure" below). Within each set, the four tracks were concatenated end-to-end with two seconds of intervening silence to stimulate their occurrence in the source playlist. The sorting of tracks into sets was made so that their average musical features were representative of the full deep focus sample (see S1 Table). The same procedure was followed for the pop hits condition, also resulting in four representative sets of four tracks each. Each participant in the deep focus and pop hit conditions heard only one of the track sets, with the specific set counterbalanced across participants. Finally, in the office noise condition, our sample comprised a single ten-minute track.

Work flow tracks were provided to us as.wav files by the music therapy app from the "anxious-to-energized" portion of their workflow playlist (sampling rate = 44.1 KHz, bit depth = 16; see Discussion for further details). Out of the twelve tracks provided, four were selected for this study based on having musical features that were broadly similar and representative of the entire set. Deep focus and pop hits tracks were purchased and downloaded as.m4a files from Apple Music (sampling rate = 44.1 kHz, bit rate = 256 kbps). For both of these conditions, we shuffled each source playlist until the first sixteen tracks were each created by a different artist, and then selected these sixteen for inclusion. The loudness of each track in the work flow condition and each set in the deep focus and pop hits conditions was normalized to -23 LUFS (Loudness Units relative to Full Scale) using the "integratedloudness.m" function from Matlab's Audio toolbox (Matlab Version R2022a; Audio toolbox Version 3.2; The Mathworks Inc.), with the result saved as a.wav file for streaming (44.1kHz/16 bit). The office noise track was generated with the following settings on the sound generator web page: "Room Tone" at 38% of maximum; "Air Co" at 50%, "Chatty Colleagues" at 89%, "Copy machine" at 66%; "Printing & Scanning" at 74%, "Office Noises" at 100%, "Keyboards & Mouse" at 58%, "Keyboards" at 73%, "Writing" at 68%; and "Office Clock" at 45% (these settings were determined by ear with the goal of simulating a typical open office environment; see supplementary materials S5 Audio for the recording used for this condition). The resulting audio stream was recorded for ten minutes as a.wav file (sampling rate = 44.1 KHz, bit depth = 16) using Audio Hijack software [62]. The loudness of the resulting track was set at -33 LUFS (i.e., 10 dB quieter than the music, with the result also saved as a.wav file for streaming (44.1kHz/16 bit).

An objective analysis of musical features for the music used in this study is shown in Table 1. This analysis was conducted using MIRtoolbox Version 1.8.1. [63, 64] in Matlab. We chose a list of 23 features designed to capture rhythmic, tonal, spectral, and dynamic aspects of the musical stimuli that are broadly associated with ratings of emotions expressed in music [65–67] but note that the best approach to feature-based music description remains a matter of debate [68]. Explanations of each feature and how they were extracted are provided in the Supplementary Materials. See S1 Table for specific track names, arrangement of tracks into sets (for deep focus and pop hits), track-by-track break downs of musical features, and the results of a clustering/silhouette analysis indicating that the work flow and deep focus tracks are generally well separated into two different groups.

**Procedure.** The experiment was conducted online. An overview of the experimental procedure is shown in Fig 1. Participants first read the informed consent. Written consent was obtained online. To do so, participants clicked on an "accept" button after reading the consent form. Participants then filled out a brief demographic survey (age, education level, and gender) and then completed three well-validated questionnaires relevant to music and mental health: the Goldsmith Musical Sophistication Index (Gold-MSI) as a continuous measure of musical

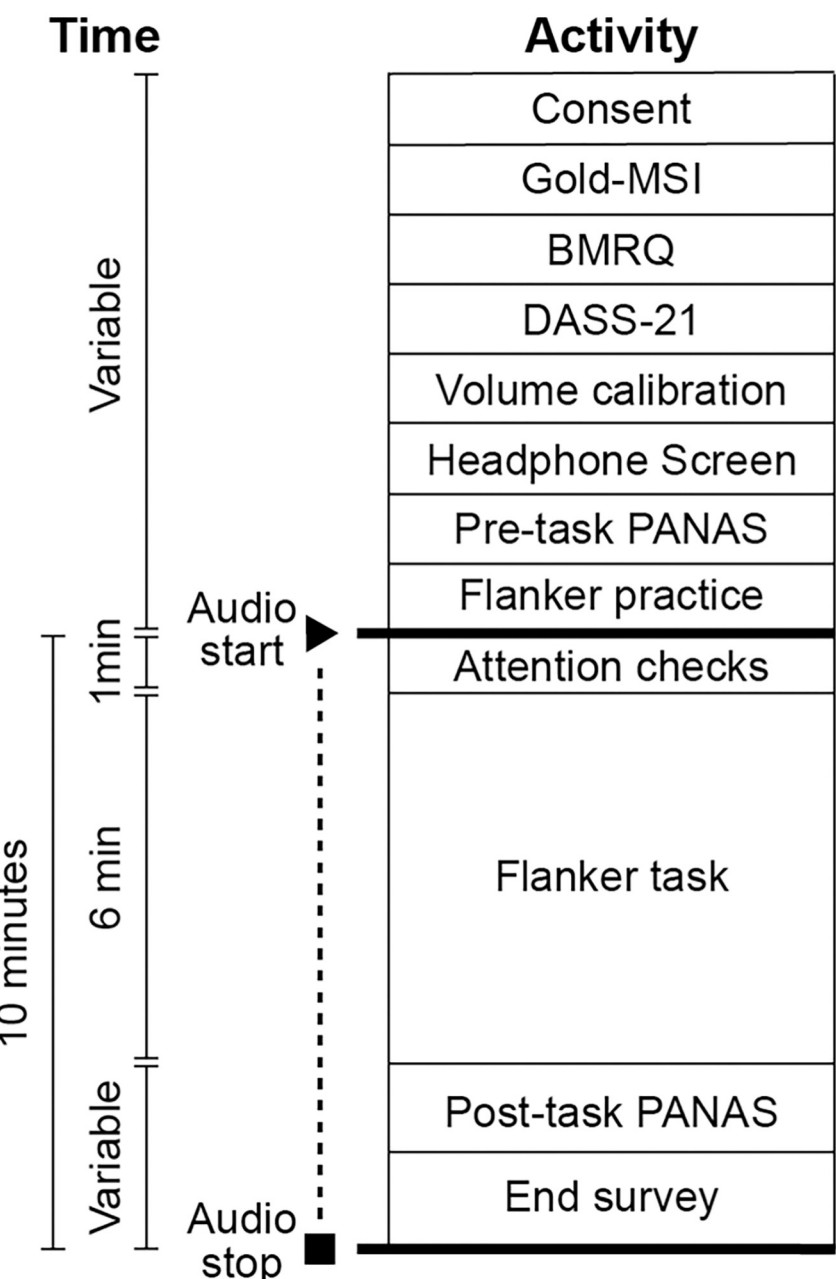

**Fig 1. Overview of the experimental procedure.**

skills and expertise [69], the Barcelona Music Reward (BMRQ) as a continuous measure of sensitivity to musical reward [70], and the 21-question version of the Depression, Anxiety, and Stress Scale (DASS-21), which measures core components of basal psychological distress [71]. To ensure data quality, all questionnaires included an item that served as an attentional check (e.g., *Please, select the option "Agree"*). Additionally, participants could not advance to the next questionnaire unless all answers had been provided.

After completing the baseline questionnaires, participants were asked to adjust their volume to a comfortable level while listening to a noise stimulus calibrated to the experimental stimuli.

Participants were then asked to record the approximate volume, as represented by their computer's operating system, by matching it to a visual analogue scale ranged between 0 and 1. After volume calibration, each participant underwent a headphone screen to determine if they were wearing functioning headphones as instructed. This screen, described in [72], is based on correctly identifying the occurrence of Huggins pitches in noise stimuli. These are very difficult to hear unless the left and right audio channels are presented dichotically.

After volume calibration and the headphone screen, the pre-task mood state of participants was measured using the Positive And Negative Affect Scales (PANAS; [73]), a well-validated measure of acute emotional status that has been successfully used to assess music-induced mood changes in pre/post listening designs like that used here [16, 74]. It comprises ten adjectives describing positive affect (e.g., "interested", "enthusiastic", etc.) and ten describing negative affect (e.g., "irritable", "upset", etc.), each rated on a five-point Likert scale. After the pre-task PANAS, participants read brief instructions on how to complete the flanker task and completed a series practice trials (detailed below). Playback of audio track or set started after completion of the practice trials. For the first minute of audio playback, on-screen text instructions "Stay tuned", "Focus on the sounds", "Enjoy the sounds", or "Listen to the sounds" were presented, allowing familiarization in the absence of an experimental task. During this period, participants were occasionally prompted with brief attention checks to ensure that they were still engaged. These consisted of responding by pressing an arrow key on the keyboard (left, right, up, or down) to describe the direction of an arrow image presented at the center of the screen. Each attention check prompt lasted just 1.5 seconds at maximum, with responses accepted anywhere within a two second window after presentation onset. To prevent anticipation, the time interval between attention checks was varied according to the formula, interval + 2—RT, where "interval" was selected (without replacement) from the set [10, 14, 16] and "RT" was the participant's RT to the previous attention check. The on-screen text between checks was randomly selected from the list provided above. After approximately one minute (depending on the intervals between attention checks), the flanker task was initiated.

On each trial of the flanker task, participants were presented with a stimulus consisting of a central right- or left-pointing arrow flanked by arrows pointing in the same direction (congruent condition), the opposite direction (incongruent condition), squares (neutral condition), or crosses (no-go condition). The task was to respond to the direction of the middle arrow by pressing the corresponding arrow key on the keyboard as soon as possible, except in trials with flanking crosses, in which case the correct response was to refrain from responding (see Fig 2). Each trial started with a black fixation dot presented in the middle of the screen for two seconds, followed by a flanker stimulus, which lasted for a maximum of two seconds. Responses were allowed any time after presentation of the stimulus within a three second window. RTs for each trial were computed from the beginning of stimulus presentation. The 24 practice trials comprised six trials per flanker condition presented in randomized order with feedback ("Correct!" or "Incorrect") given immediately after a response, or after the three second window if no response was made. The structure of the test trials was the same except no feedback was provided. Each participant completed 72 test trials (18 per condition, half with the center arrow pointing right, half with it pointing left). Additionally, test trials were made to occur exactly 5 seconds apart by introducing variable duration intervals after participant responses such that RT + interval was always equal to three seconds (plus two seconds for the pre-stimulus fixation dot equals five seconds of total trial time). Standardizing trial duration allowed us to fix the duration of the flanker task at 6 minutes across all participants, regardless of differences in RT.

After the flanker task (approximately seven minutes after initiating audio playback), participants completed a post-task PANAS, allowing us to compute changes in mood. Finally,

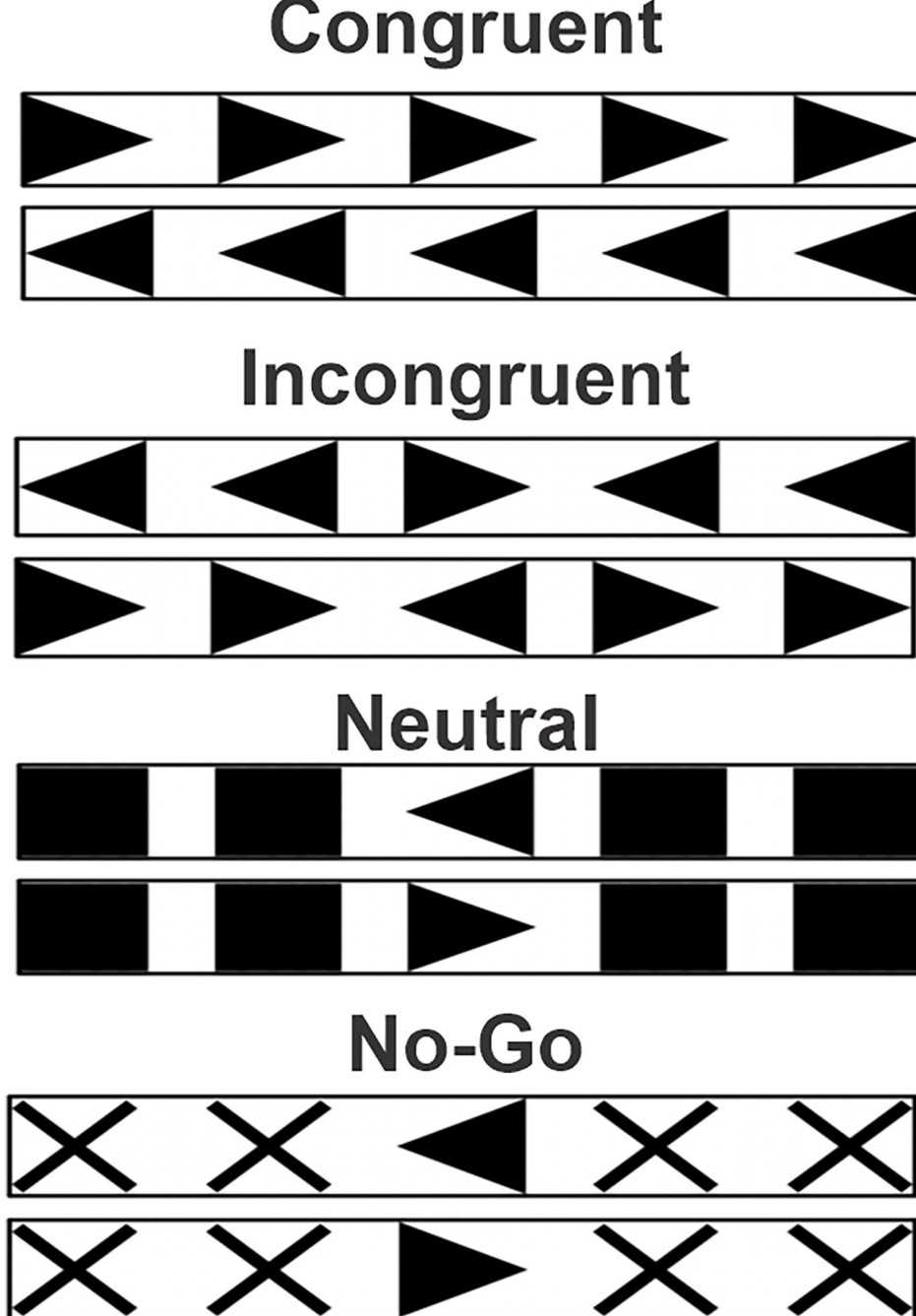

**Fig 2. Flanker task conditions.** Note that there are two trial types per condition, central arrow pointing left or right.

participants completed a short end survey about their listening experience. This survey examined subjective perceptions of pleasure ("How much did you like the soundtrack that you heard?"), groove ("How much did the soundtrack make you want to move? For example, by tapping your foot, bobbing your head, or rocking back and forth"), and familiarity ("How familiar are you with the soundtrack you heard?"). For the office noise condition, we changed the word soundtrack for "audio". These final questions were answered using a slider with

values ranging between 0 and 1. Audio playback was stopped precisely ten minutes after initiation. The experiment was terminated at this point unless the end survey was not yet complete. In cases where the end survey was completed before ten minutes of audio had elapsed, an on-screen message instructed participants to continue listening for the remaining time.

## Statistical analysis

To determine whether participants assigned to the four audio conditions were balanced in terms of demographic variables (age, gender, level of education), musical skills and expertise (Gold-MSI), sensitivity to musical reward (BMRQ), and acute psychological distress (DASS-21), we used Bayes Factors (BFs) as implemented in JASP using default priors [75–78]. We also tested whether there were differences between the groups for the self-reports of volume. We report $BF_{01}$, which reflects the probability of the data given H0 relative to H1, which in our case quantifies the strength of the evidence supporting the hypothesis that the groups are equal on the indicated variables (H0), relative to the strength of the evidence supporting the hypothesis that the groups are different (H1). For example, a $BF_{01} = 4$ can be interpreted as the data being 4 times more likely under H0 than under H1 [79]. For continuous variables (age, GOLD-MSI, BMRQ, and DASS-21) we used Bayesian one-way ANOVAs to compare the groups in the different audio conditions. For categorical variables (education, gender) we used Bayesian Contingency Tables with independent multinomial sampling to compare conditions, as participants were randomized to one of four audio conditions with the aim of collecting approximately the same number of participants in each condition [80].

To test the effects of audio condition on mood (PANAS), we used a single one-way between-participants ANOVAs with four levels (work flow, deep focus, pop hits, office noise) for each dependent variable, implemented in JASP. For an overall assessment of mood effects, we examined total change in PANAS scores, calculated as change-in-PANAS-positive + (-1) *change-in-PANAS-negative. To assess effects on positive and negative affect separately, we subsequently analyzed changes in the PANAS positive and negative scales. For significant effects of audio condition, post-hoc t-tests were used to compare each possible pairing of groups, with Tukey correction for multiple comparison. One-way between-participants ANOVAs with four levels (work flow, deep focus, pop hits, office noise) followed by post-hoc t-tests with Tukey correction were used to analyze the self-reported musical familiarity, pleasure, and groove ratings made during the end survey. Partial eta$^2$ ($\eta_p^2$) and Cohen's d were used as measures of effect size.

To test the effects of the audio condition on performance, we conducted several analyses. First, we examined accuracy, using the number of correct trials in the Flanker Task as the dependent variable, and a mixed between-within 4 x 4 repeated measures ANOVA with audio condition (work flow, deep focus, pop hits, office noise) as the between-subjects factor and flanker condition (congruent, incongruent, neutral, no-go) as the within-subjects factor. For significant effects, we used post-hoc tests with Holm correction for multiple comparisons. Second, we also examined reaction times (RTs). RTs were only analyzed for correct trials. We opted for linear mixed models rather than ANOVAs for analyzing RTs [81, 82]. We employed the lme4 package [83] within R (version 4.2.0) and RStudio (version 2022.02.2.485) for this analysis. We discarded any trial with an RT faster than 150ms, which was deemed too fast to be a real response (typical reaction times vary between 400–500 ms; 46,47). For the final sample of participants (N = 196; see Results) only four trials were discarded for this reason, out of a total of 10,152 correct trials across all participants and conditions.

For linear mixed modeling, we first generated a null model with a random intercept for participant but no experimental effects. We then defined the minimal model as that containing a

random intercept effect for participant, and fixed factors effects for: i) audio condition (work flow, deep focus, pop hits, office noise) and ii) flanker condition (congruent, incongruent, and neutral; no-go trials do not have RTs); iii) the BMRQ, to control for individual variation in sensitivity to musical reward (this scale is known to correlate with neurophysiological and behavioral responses to music; [27, 84, 85]); iv) the Music Training subscale of the Gold-MSI, to control for individual variation in musical skills and expertise; and v) the three subscales of the DASS-21 (Depression, Anxiety, and Stress), to consider effects of basal psychological distress. The minimal model included no interactions. Next, we generated a set of models by adding trial number to assess the effect of time on performance (note that trial number is directly proportional to time, since each trial was spaced exactly 5 seconds apart; see "Procedure" above) and modeling all possible interactions between audio condition, flanker condition, and trial number. Finally, we compared the Akaike information criterion (AIC) of each model to determine which model explained the most variance in the data with the most efficient effect structure. In accord with established standards for comparing AICs, we considered a model superior to another if its AIC was two or more points smaller [34, 86, 87]. Each effect in the best model was further examined using Type III Wald chi-square tests, as implemented in the *car* package. Pairwise contrasts for significant effects were carried out using the *emmeans* package and Tukey correction for multiple comparisons. Predicted effects were plotted using the *ggpredict* package. Data is available as supplementary material files. Analysis code followed standard pipelines in R and JASP.

## Results

### Participants

Out of the total 280 participants recruited, 84 were excluded for the following reasons: 18 had musical anhedonia, as defined by a score of less than 63 on the BMRQ [70, 84], 32 failed the headphone screener, 17 responded only in fewer than 50% of the flanker test trials (i.e., fewer than 26 responses on the 54 congruent, incongruent, or neutral trials in which a response was expected), 15 failed one or more of the attention checks in the baseline questionnaires or during the first minute of audio playback, and 2 responded on more than 80% of the flanker test trials with implausibly fast RTs (i.e., <150 ms). The final sample thus consisted of 196 participants (mean age = 38.14 years, SD = 9.72 years; 80 female, 116 male; 160 with a college degree or higher, 36 with a high-school diploma). This included 50 assigned to the work flow condition, 50 to deep focus, 49 to pop hits, and 47 to office noise. The attrition rate of 30% is consistent with other online behavioral studies using participants from Amazon Mechanical Turk completed by the authors, including studies with similarly complex tasks and auditory stimuli [50, 51, 53, 87].

### Group characterization

Bayes factors show that the groups were well balanced (see S1 Fig in S1 File) in terms of demographic variables (age $BF_{01}$ = 30.00; gender $BF_{01}$ = 56.94; education $BF_{01}$ = 83.46), sensitivity to musical reward (BMRQ: total $BF_{01}$ = 35.98; seeking $BF_{01}$ = 15.67; emotion $BF_{01}$ = 11.11; mood $BF_{01}$ = 24.09; sensorimotor $BF_{01}$ = 19.44; social: $BF_{01}$ = 13.05), basal mental health status (DASS-21: depression $BF_{01}$ = 15.26; anxiety $BF_{01}$ = 5.11; stress $BF_{01}$ = 9.40), and computer volume ($BF_{01}$ = 13.80). For musical training the $BF_{01}$ was 2.11. This lower BF is driven by a higher score displayed by the participants in the pop hits group (see S1D Fig in S1 File). However, there was no significant effect of group of musical training ($F(3,192)$ = 2.38, p = 0.07) with all post-hoc tests comparing each group score against each other showing no significant differences (all corrected ps>0.092). Music genre preferences varied within groups but were

comparable across them. Among the four groups, the two most popular genres were Pop and Rock (see S2 Table in S1 File). An analysis of the genres selected as favorites by at least 10 participants—namely Rock, Pop, Classical, Rap, and Jazz—revealed no significant differences in their distribution across the groups ($BF_{01} = 1081$).

### Effect of audio condition on mood

There was a significant effect of audio condition on the total change in PANAS score ($F_{3,192} = 11.22$, p<0.001, $\eta_p^2 = 0.149$; Fig 3A), with post-hoc tests showing that participants in the work flow condition experienced greater improvements in mood than those in the deep focus condition ($t_{98} = 4.15$, $p_{holm}$<0.001, d = 0.83), pop hits condition ($t_{97} = 5.03$, $p_{holm}$<0.001, d = 1.01), and office noise condition ($t_{95} = 4.79$, $p_{holm}$<0.001, d = 0.97). No other between-group comparisons were significant (all ps>0.080). Furthermore, one-sample t-tests applied to each condition indicated that work flow was the only audio condition in which the total change in PANAS score from before to after the flanker task was significantly different than zero (work flow, $t_{49} = 7.0$, p<0.007, d = 0.99, this p-value survives Bonferroni correction for the 4 tests computed; focus, $t_{49} = -0.879$, p = 0.38, d = 0.124; pop hits, $t_{48} = -1.56$, p = 0.12, d = 0.224; office noise, $t_{46} = -1.55$, p = 0.12, d = 0.227). At the individual level, 76% of participants in the work flow condition reported an increase in total PANAS score, reflecting a net improvement in mood; the same percentage was less than half for each of the other three conditions (44% for deep focus, 36.7% for pop hits, and 36.17% for office noise). Similar results were obtained when separately analyzing changes in the PANAS positive scale ($F_{3,192} = 8.31$, p<0.001, $\eta_p^2 = 0.115$)—with participants in the work flow condition experiencing greater increases in positive affect than those in the deep focus ($t_{98} = 3.62$, $p_{holm} = 0.002$, d = 0.72), pop hits ($t_{97} = 4.23$, $p_{holm}$<0.001, d = 0.85), and office noise conditions ($t_{95} = 4.22$, $p_{holm}$<0.001, d = 0.85)—as well

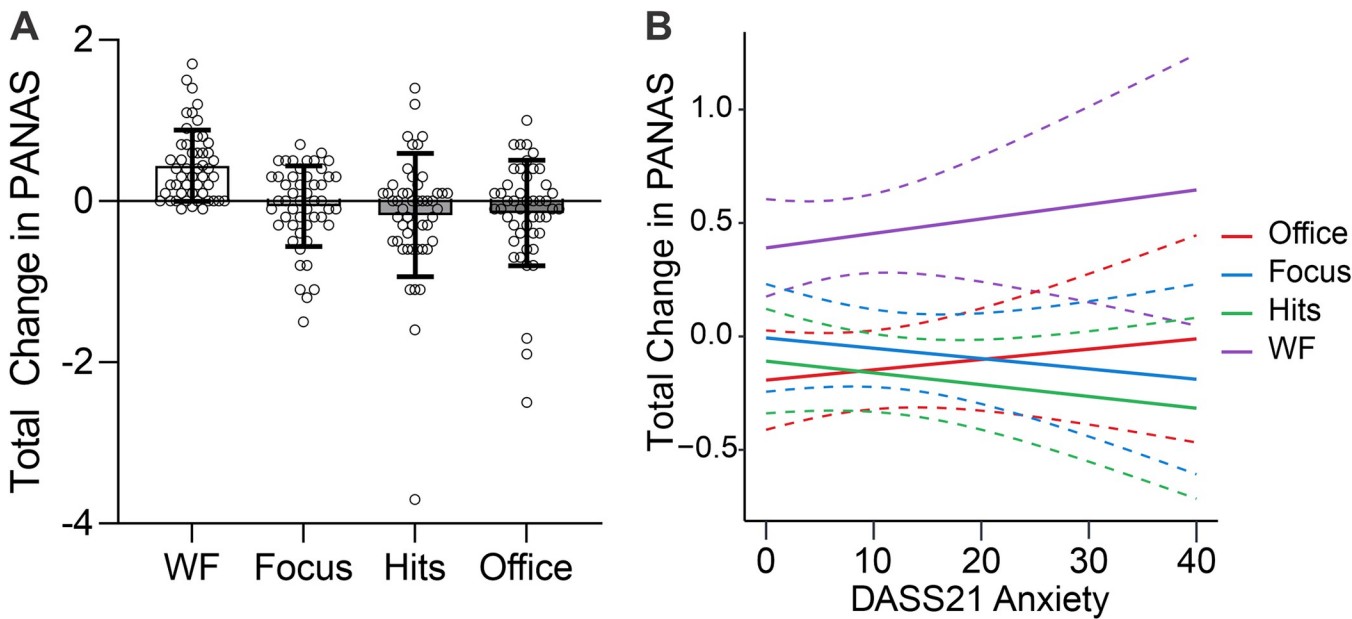

**Fig 3. Effects of audio condition on mood. A.** Participants in the work flow (WF) condition exhibited greater improvements in mood than participants in the deep focus (Focus), pop hits (Hits), and office noise (Office) conditions (all ps<0.001, corrected for multiple comparisons). Bar plots show mean with standard deviation. **B.** Exploratory analysis indicating that mood improvement in the work flow condition was independent of individual variation in basal levels of anxiety as assessed with the DASS-21 (i.e., audio condition did not interact with the DASS-21 anxiety subscale). The same was true for the DASS-21 depression and stress subscales, but only results for anxiety are shown here. Solid lines show predicted values; dashed lines show 95% confidence intervals. For reference, DASS-21 Anxiety scores 0–7 are considered normal, 8–9 are considered mild, 10–14 moderate, 15–19 severe, and 20+ extremely severe.

as the PANAS negative scale ($F_{3,192}$ = 6.40, p<0.001, $\eta_p^2$ = 0.091)—with participants in the work flow condition experiencing greater decreases in negative affect than those in the deep focus ($t_{98}$ = 3.06, $p_{holm}$ = 0.010, d = 0.61), pop hits ($t_{97}$ = 3.92, $p_{holm}$<0.001, d = 0.79), and office noise conditions ($t_{95}$ = 4.79, $p_{holm}$ = 0.003, d = 0.71).

Given the significant main effect of audio condition on mood, an exploratory analysis was performed to determine whether the effect of the work flow condition on mood interacted with our basal measures of psychological distress. Using R, we computed three simple linear models predicting total change in PANAS score as a function of the interaction between audio condition and each of the DASS-21 subscale scores (e.g., Total change in PANAS score = AudioCondition*DASS-21Anxiety). We used F-tests implemented in the function *anova* to compare each linear model to a version of itself specified without the interaction (e.g., Total change in PANAS score = AudioCondition+DASS-21Anxiety). These tests indicated no significant interactions between audio condition and anxiety (p = 0.589; see Fig 3B), depression (p = 0.351), or stress (p = 0.544), suggesting that beneficial effects of the work flow music on mood during task performance observed here are robust to individual variation in basal levels of depression, anxiety, and stress.

The results of the end survey are shown in Fig 4 (these data were missing for two participants in the work flow condition, three in the deep focus condition, five in the pop hits condition, and three in the office noise condition). There was a significant effect of audio condition on pleasure ($F_{3,179}$ = 12.52, p<0.001, $\eta_p^2$ = 0.17), with post-hoc tests showing that participants in the office noise condition reported liking the audio less than those in the work flow ($t_{90}$ = 5.01, $p_{holm}$<0.001, d = 1.04), deep focus ($t_{89}$ = 5.54, $p_{holm}$<0.001, d = 1.16), and pop hits conditions ($t_{86}$ = 2.92, $p_{holm}$ = 0.016, d = 0.62). In addition, participants in the pop hits condition reported liking the audio less than those in the deep focus condition ($t_{89}$ = 2.57, $p_{holm}$ = 0.033, d = 0.54), although the difference with the work flow condition was not significant ($t_{90}$ = 2.02, $p_{holm}$ = 0.088, d = 0.423). There was also a significant effect of audio condition on self-reported ratings of groove ($F_{3,179}$ = 18.85, p<0.001, $\eta_p^2$ = 0.24), with post-hoc tests showing that participants in the work flow and pop hits conditions reported that the audio stimulated greater desire for movement than those in the deep focus condition (work flow vs. deep focus: $t_{93}$ = 2.78, $p_{holm}$ = 0.017, d = 0.57; pop hits vs. deep focus: $t_{89}$ = 2.80, $p_{holm}$ = 0.017, d = 0.58). Not surprisingly, participants in the office noise condition reported lower groove scores than those

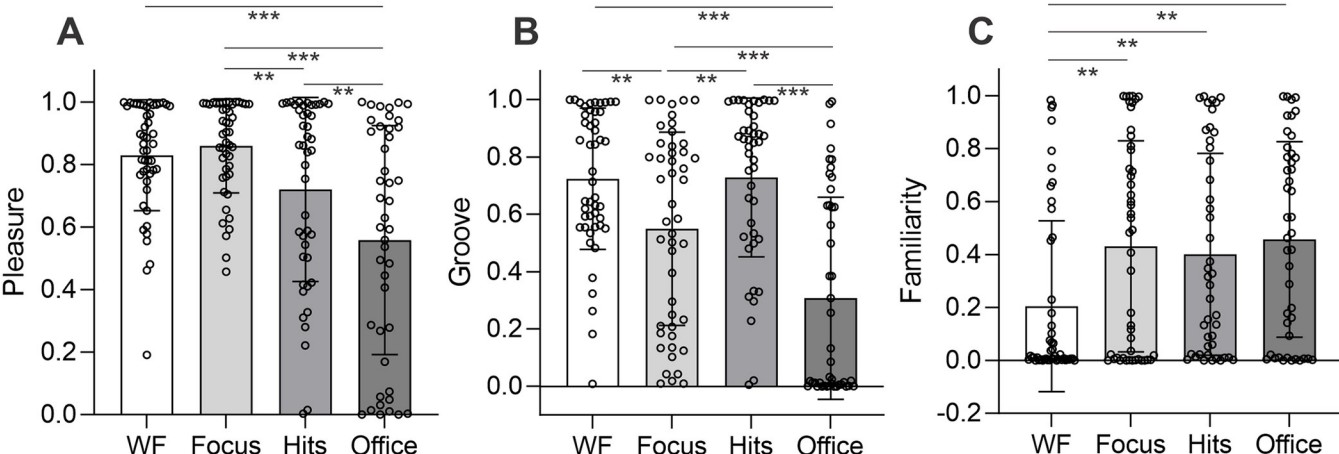

**Fig 4. End survey results.** Self-reported pleasure (**A**), groove (**B**), and familiarity (**C**) associated with the audio in the work flow (WF), deep focus (Focus), pop hits (Hits), and office noise (Office) conditions. Bars show mean with standard deviation. *** p<0.001, **p<0.05.

in each of the other three audio conditions (work flow: $t_{90}$ = 6.53, $p_{holm}$<0.001, d = 1.36; pop hits: $t_{86}$ = 6.46, $p_{holm}$<0.001, d = 1.37; deep focus: $t_{89}$ = 3.76, $p_{holm}$ = 0.001, d = 0.79). Finally, there was a significant effect of audio condition on self-reported ratings of familiarity ($F_{3,179}$ = 4.55, p = 0.004, $\eta_p^2$ = 0.071), with participants in the work flow condition reporting that the audio was less familiar than those in the deep focus ($t_{93}$ = 2.99, $p_{holm}$ = 0.016, d = 0.61), pop hits ($t_{90}$ = 2.54, $p_{holm}$ = 0.048, d = 0.53), and office noise conditions ($t_{90}$ = 3.22, $p_{holm}$ = 0.008, d = 0.68).

Given these significant differences in pleasure and familiarity between audition conditions, both of which are generally related to effects of music on mood [88], we performed a further exploratory analysis to determine if mood changes were related to pleasure or familiarity. We computed two simple linear models predicting total change in PANAS score as a function of the interaction between audio condition and pleasure or familiarity (e.g., Total change in PANAS score = AudioCondition*Pleasure). We again used F-tests implemented in the function *anova* to compare each linear model to a version of itself specified without the interaction (e.g., Total change in PANAS score = AudioCondition+Pleasure). No significant interactions were observed between audio condition and pleasure (p = 0.108) or audio condition and familiarity (p = 0.745), suggesting that differences in mood change between audio conditions are not a result of associated differences in either of these variables. That said, these analyses did indicate a significant main effect of pleasure on total change in PANAS score ($t_{178}$ = 7.2, p<0.001; see S5 Fig in S1 File), suggesting that the more participants liked the audio in any condition, the more their mood improved from before to after the flanker task. We did not observe a main effect of familiarity on total change in PANAS score ($t_{175}$ = 0.042, p = 0.96).

## Effect of background audio on performance

The results of the flanker task accuracy analysis are shown in S2 Fig in S1 File. A mixed between-within repeated measures ANOVA showed a main effect of flanker condition on accuracy ($F_{3,576}$ = 38.46, p<0.001, $\eta_p^2$ = 0.167), with post-hoc tests showing that participants made significantly more errors in the no-go condition than in the congruent ($t_{195}$ = 9.62, $p_{holm}$<0.001, d = 0.82), incongruent ($t_{195}$ = 5.81, $p_{holm}$<0.001, d = 0.49), and neutral conditions ($t_{195}$ = 8.93, $p_{holm}$<0.001, d = 0.76). In addition, participants made significantly more errors in the incongruent condition than in the congruent ($t_{195}$ = 3.80, $p_{holm}$<0.001, d = 0.32) or the neutral conditions ($t_{195}$ = 3.11, $p_{holm}$ = 0.004, d = 0.26). These results replicate standard accuracy effects for the flanker task [46, 47]. However, the effect of audio condition was not significant ($F_{3,192}$ = 1.07, p = 0.36, $\eta_p^2$ = 0.016), nor was its interaction with flanker condition ($F_{9,576}$ = 6.81, p = 0.65, $\eta_p^2$ = 0.012). This implies that flanker task accuracy was not significantly modulated by the different types of music tested in this study.

We next examined flanker task RTs (or speed). The different candidate models used to assess the effect of audio condition on flanker task RT are shown in Table 2. Examining the best model [*FlankerCondition + AudioCondition*TrialNumber + BMRQ + GOLD-MSI + DASS21 + (1|ID);* see S3 Table in S1 File for a full list of parameter estimates with 95% confidence intervals)], as determined by AIC comparisons, showed a main effect of flanker condition ($\chi_2^2$ = 361.62, $p < 0.001$). Post-hoc tests showed that participants reacted faster in the congruent compared to incongruent (congruent minus incongruent: -89.5±4.97 ms, $t_{9961}$ = -18.01, $p_{tukey}$<0.001) and neutral conditions (congruent minus neutral: -17.4±4.89 ms, $t_{9959}$ = -3.55, $p_{tukey}$ = 0.001), and slower in the incongruent compared to neutral conditions (incongruent minus neutral: +72.1±4.97 ms, $t_{9960}$ = 14.48, $p_{tukey}$<0.001; see S3 Fig in S1 File). These results are also consistent with typical flanker task performance [46, 47].

**Table 2. Candidate linear mixed models for flanker task RT.**

| Linear mixed model predicting RT | $K_i$ | AIC | $\Delta(AIC)$ |
|---|---|---|---|
| **FlankerCondition + AudioCondition\*TrialNumber + BMRQ + GOLD-MSI + DASS21 + (1\|ID)** | **17** | **137434.8** | **0.00** |
| FlankerCondition + AudioCondition + TrialNumber + BMRQ + GOLD-MSI + DASS21 + (1\|ID) | 14 | 137445.0 | 10.23 |
| FlankerCondition\*TrialNumber + AudioCondition + BMRQ + GOLD-MSI + DASS21 + (1\|ID) | 16 | 137446.0 | 11.22 |
| FlankerCondition + AudioCondition + BMRQ + GOLD-MSI + DASS21+(1\|ID) | 13 | 137448.0 | 13.19 |
| FlankerCondition\*TrialNumber\*AudioCondition + BMRQ + GOLD-MSI + DASS21 + (1\|ID) | 31 | 137448.9 | 14.09 |
| FlankerCondition\*AudioCondition + TrialNumber+ BMRQ+GOLD-MSI +DASS21 + (1\|ID) | 20 | 137449.2 | 14.42 |
| FlankerCondition\*AudioCondition + BMRQ + GOLD-MSI + DASS21 + (1\|ID) | 19 | 137452.2 | 17.37 |
| Empty Model | 3 | 137881.8 | 446.97 |

All models included random intercepts for participants *(1|ID)*.

\* indicates an interaction. DASS21 stands for three fixed factors: DASS21Anxiety+DASS21Depression+DASS21Stress. $K_i$ = the number of estimated parameters in each model. AIC = corrected Akaike information criterion. $\Delta(AIC)$ = difference between AIC for each model and the best model. The best model appears in in bold.

Intriguingly, there was a main effect of basal anxiety levels on flanker task RT, with participants who scored higher on the DASS-21 anxiety subscale responding slower overall (β = 18.22, i.e., for each one-point increment in DASS-21 anxiety score, RTs increased by 18 ms; $\chi^2_1$ = 30.82, $p<0.001$; see S4A Fig in S1 File). A main effect was also observed for baseline sensitivity to musical reward, with participants who scored higher on the BMRQ responding faster overall (β = -6.02, i.e., for each one-point increment in BMRQ score, RTs decreased by 6 ms; $\chi^2_1$ = 7.79, $p = 0.005$; see S4B Fig in S1 File]. This result remained significant when excluding data from participants in the office noise condition ($\chi^2_1$ = 5.45, $p = 0.019$). Finally, there was a significant interaction between audio condition and trial number ($\chi^2_3$ = 16.26, $p = 0.001$; see Fig 5A). Post-hoc tests showed that participants in the work flow condition became faster over time, regardless of flanker condition with the rate of decrease in RT (work flow slope: β = -0.421) being significantly steeper than in each of the other conditions (deep focus slope: β = 0.615, difference in slope with work flow = 1.03, $t_{9959}$ = 3.80, $p_{tukey}<0.001$; pop hits slope: β =

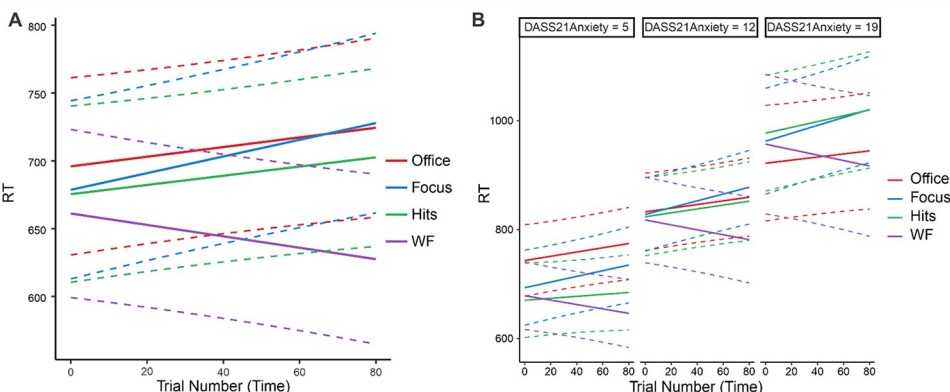

**Fig 5. Effect of audio condition on flanker task RT over time. A.** RT slopes with increasing trial number (time) for each audio condition, as predicted by the best RT model (see Table 2). Participants in the work flow (WF) condition performed significantly faster over time, as compared to those in the deep focus (Focus), pop hits (Hits), and office noise (Office) conditions (see main text for stats). **B.** Participants in the work flow condition performed faster over time regardless of basal anxiety status, as indicated by the similarity of RT slopes with increasing trial number for DASS-21 anxiety scores corresponding to "normal" (score = 5), "moderate" (score = 12), and "severe" (score = 19) levels. That is, the interaction between audio condition, trial number, and basal anxiety status was non-significant (see main text for stats). Dashed lines represent 95% confidence intervals.

0.340, difference in slope with work flow = 0.76, $t_{9959}$ = 2.79, $p_{tukey}$ = 0.027; office slope: β = 0.356, difference in slope with work flow = 0.77, $t_{9959}$ = 2.83, $p_{tukey}$ = 0.024). This indicated a significant effect of work flow music on performance (i.e., faster responses over time, with similar flanker accuracy). That is, despite similar levels of accuracy, participants in the work flow condition reacted significantly faster to flanker stimuli over time.

Given this effect of work flow music on RT over time, combined with the fact that RT was negatively impacted by high levels of basal anxiety, an exploratory analysis was performed to assess the potential relationship between these effects. Specifically, we aimed to test whether the effect of work flow music on RT over time was dependent on anxiety level. Accordingly, using R, we computed a linear mixed model to predict RT that was the same as the best model described above (see also Table 2) except that it also included the three-way interaction between audio condition, trial number, and DASS21 anxiety subscore [i.e., FlankerCondition + AudioCondition*TrialNumber*DASS21Anxiety + DASS21Depression + DASS21Stress + BMRQ + GOLD-MSI + (1|ID)]. The results showed that this three-way interaction was not significant ($\chi^2_3$ = 3.01, $p$ = 0.389). This suggests that the observed effect of work flow music on task performance over time was independent of anxiety severity as assessed by the DASS-21 (see Fig 5B), despite the overall slower performance of individuals with high levels of anxiety (see S4A Fig in S1 File).

In a final exploratory analysis, and for the work flow condition only, we assessed the relationship between the significant observed effects on mood and task performance. Specifically, we aimed to test whether the magnitude of mood improvement was related to the magnitude of RT decrease over time. To do this we computed a linear mixed model predicting RT as a function of fixed effects for trial number, total change in PANAS score, and their interaction, as well as a random intercept effect for participant [i.e., RT ~ TrialNumber*TotalPANAS+ (1|ID)]. The results (see S4 Table in S1 File for a full list of parameter estimates with 95% confidence intervals) showed that the interaction term was significant ($\chi^2_1$ = 4.96, p = 0.0258), suggesting that for participants in the work flow condition (N = 50), larger improvements in mood were associated with increasingly faster performance over time (Fig 6).

## Discussion

In this study we investigated the effect of two acoustically distinct forms of music advertised to support attentional focus and concentration on mood and performance during a flanker task. The "work flow" music we tested was characterized by strong rhythm with simple tonality, broadly distributed spectral energy below ~6000 Hz, and moderate dynamism. By contrast, the "deep focus" music that we tested was more minimalistic, with similarly simple tonality, but weaker rhythm, lower and more restricted spectral energy, and more reserved dynamism (see Table 1 and S1 Table). The effects of these types of music were compared with those of popular music and office noise, representing mainstream musical stimulation and the acoustic background in a naturalistic social working environment, respectively. The experiment was conducted between-subjects, with approximately 50 participants per condition. Groups were well-balanced in terms of basic demographics, musical training, sensitivity to musical reward, basal mental health status, and stimulus volume.

Regarding mood, the results showed that only the work flow music had a significant effect, driving improvements in overall mood from before to after the flanker task (see Fig 3A), underpinned by increases in positive affect and decreases in negative affect. While there was not a significant interaction between audio condition and pleasure, there was a main effect of pleasure on total change in PANAS score, with greater improvements in mood being associated with greater music liking across conditions. Importantly for applications of music in

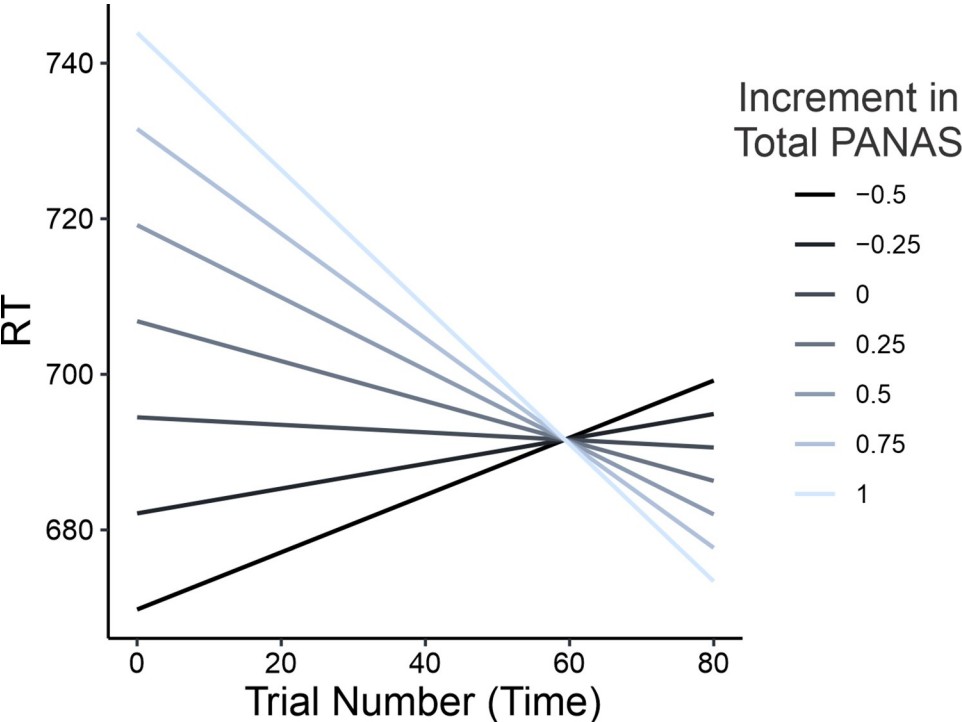

**Fig 6. Relationship between effects on mood and performance in the work flow condition.** Each line shows the model prediction for RT over time for a different change in total PANAS score (from -0.25 to 1). For participants who listened to work flow music, larger improvements in mood (black to blue gradient) were associated with increasingly rapid responses on flanker trials with increasing trial number (i.e., steeper slopes; p = 0.0258).

mental health and wellness, the observed effect of work flow music on mood was independent of individual variation in baseline levels of self-reported anxiety, depression, or stress over the past week, as measured by the DASS-21 (see Fig 3B). This suggests that work flow music may be effective for mood management even when people are suffering from emotional distress (approximately 46%, 30%, and 20% of the work flow sample scored in the moderate to extremely severe range for depression, anxiety, and stress, respectively). Regarding flanker task performance, the results showed that, while response *accuracy* was not affected by audio condition, response *speed* was. Specifically, we observed a significant interaction between audio condition and trial number, such that participants listening to work flow music responded more quickly over time. Importantly for applications of music in mental health and wellness, this effect was independent of individual differences in self-reported anxiety, which otherwise negatively impacted response speed. This suggests that work flow music may be useful for people losing focus due to high levels of anxiety (approximately 30% of the work flow sample scored in the moderate to extremely severe range for anxiety). Finally, in the work flow condition, effects on mood and performance were correlated such that greater improvements in mood were associated with increasingly faster responses, regardless of flanker stimulus condition (see Fig 6).

The work flow music used here was composed by professional musicians commissioned by a music therapy app for the purpose of supporting on-task functioning at work. In creating these tracks, composers were asked to follow detailed acoustic guidelines specific to the listening category "work flow" and an emotional trajectory called "anxious-to-energized". These guidelines were created by the music therapy app in consultation with co-authors D.L. B. and

C.T. D.L.B. and C.T. also reviewed each track in an iterative editorial process in which adherence to the acoustic guidelines was maintained. In broad terms, these guidelines specified that the majority of each track comprised strong rhythmic features to support groove, simple melodic and harmonic features, and an explicit avoidance of features that readily divert attentional focus (e.g., concentrated high frequency energy, highly articulated attacks, lyrics; [29, 59, 89, 90]). By contrast, the deep focus music used here can be expected to exhibit greater diversity in compositional process, having instead been deemed suitable to support on-task functioning after composition took place, by the personnel and/or algorithms that curated the source playlist.

The PANAS results showing significant changes in mood after anywhere from 7–10 minutes of listening to work flow music (see Fig 3A) add to the growing literature showing that listening to music can improve mood at very short timescales [13, 91, 92]. This raises the question of why no significant changes in mood were observed in the other music conditions. As already mentioned, no significant differences in self-reported liking were observed between work flow and deep focus music, suggesting that the absence of mood effects in the deep focus condition is not explained by the music being inherently less pleasurable. Instead, we speculate that the minimalistic nature of the deep focus music, especially its relatively low energy level (as implied by, e.g., lower spectral flux and entropy; see Table 1; [67]), may have been responsible for its failure to inspire a mood shift. Another, non-mutually exclusive possibility is related to the fact that the work flow stimuli were coherent wholes, each comprising one evolving track with a duration of approximately 10 minutes. By contrast, the deep focus tracks were only approximately three minutes each, thus requiring the combination of multiple tracks into sets to achieve the same duration (dosage) for comparison. While these sets were assembled with the intention of maintaining uniformity in musical features (see S1 Table), inevitable musical differences and inter-track transitions may have nonetheless interfered with the consistency of mood effects. It is plausible, for example, that entrainment facilitated by listening to a continuous, coherent track may have aided participants to perform the task correctly faster, whereas adapting to a dynamically changing auditory environment could have increased cognitive load and impaired performance. In the pop hits condition, interference of this kind can be expected to have been even greater, as variability between tracks was more pronounced (see S1 Table). Further, given that pop hits come from different music genres, that individual genre preferences vary widely, and that different genres can have very different mood effects [93], the mix of genres in the pop hits condition may have further precluded any consistent mood effects across listeners.

Turning to the observed effect of work flow music on task performance, our results are consistent with a large body of research showing that listening to music can have beneficial effects on various aspects of cognition including verbal learning, memory, and semantic fluency, as well as attention in some cases [33–39, 60, 94–96]. The effect of work flow music on flanker task performance observed here was a general increase in response speed that was observed across flanker stimulus conditions (i.e., congruent, incongruent, and neutral). While much research using the flanker task has traditionally aimed to index selective attention by focusing on the flanker interference effect (i.e., the difference in RT between incongruent and congruent trials; [97]) as an index of selective attention, we note that recent research assessing the effect of background music found results consistent with ours. As mentioned in the introduction, two recent studies reported that participants also showed a general increase in response speed that was independent of flanker stimulus condition while listening to "joyful" classical music [38, 39].

The results of this study appear to be in good accord with arousal-mood theory, which posits that positive effects of music listening on cognition can be understood in terms of music's

well-documented capacities to upregulate arousal level and positive affect [98]. Although we did not explicitly measure subjective perceptions of arousal, there is good reason to expect that both the work flow tracks and pop hits tracks used here were relatively arousing, while the deep focus music was not. The perception of groove is related to arousal, measured physiologically or perceptually [90, 99], and was comparable between the work flow and pop hits conditions (means = $0.724\pm0.246$ and $0.729\pm0.277$, respectively; $BF_{01} = 4.55$; see Fig 4). Perceived groove in these conditions was also significantly higher than it was in the deep focus condition (mean = $0.549\pm0.337$; see Fig 4). Moreover, the work flow and pop hits tracks were relatively high in musical features closely associated with perceived arousal, such as spectral flux, spectral entropy, pulse clarity, and fluctuation maximum (see Table 1; [65, 67]), whereas the deep focus tracks exhibited lesser values for these features. Regarding positive affect, only the work flow tracks increased positive affect, as measured by the PANAS survey. That said, both the work flow and deep focus tracks were reasonably well-liked (mean pleasure = $0.829\pm0.177$ for work flow, $0.860\pm0.150$ for deep focus; $BF_{01} = 3.25$), and relatively more so than the pop hits tracks, which were rated lower on average (mean = $0.720\pm0.294$; Fig 4), and more variably overall (see error bars). In the context of the arousal-mood theory, the beneficial effects of work flow music on performance speed can thus be rationalized by its capacity to stimulate both high arousal and positive affect *simultaneously*. By contrast, the other music conditions may have failed to modulate response speed because of lesser capacities to do so: the deep focus tracks were pleasurable but not arousing; the pop hits tracks were arousing but not consistently pleasurable.

Additional support for the arousal-mood theory comes from our exploratory analysis showing that for participants in the work flow condition, greater improvements in mood were associated with more rapid improvements in response speed over time (see Fig 6). These results are of particular importance when paired with our finding that individuals with higher anxiety levels had slower RTs in the flanker task (see S4A Fig in S1 File). Anxiety is well-known to hinder cognition [100–103], including selective attention [104]. This raises the possibility that, under some circumstances, music modulates cognition via emotional effects that downregulate anxiety and its detrimental effect on cognition, leading to an increase in performance speed. In this context, we also note our finding that individuals with higher sensitivity to musical reward had faster RTs in the flanker task (see S4B Fig in S1 File), which is consistent with some previous research showing that patients with higher sensitivity to musical reward benefit the most from music-based interventions [105]. If high sensitivity to musical reward confers heightened sensitivity to music's emotional effects, such individuals may also benefit the most from using music to improve task performance. Finally, in considering our results in the context of arousal-mood theory, it should be noted that the latter was developed to explain cognitive effects driven by listening to music *before* performing a particular task [98], whereas our results (and those of other recent studies, e.g., [38, 39]) primarily concern music listening *during* task performance. These results suggest that the theory may hold equally well in both cases.

Our study suffers from a few limitations. First, data was collected online and while multiple attentional checks were implemented (including a test ensuring that participants were wearing headphones), online experiments are inevitably less well-controlled than laboratory studies. Second, the study involved four different groups of participants, and while we confirmed that these groups were comparable in terms of basic demographic, music-related, and mental health variables, there may have been relevant group-level differences that our surveys might not have captured. Third, we did not explicitly measure subjective impressions of arousal, which may (or may not) have provided more direct support for our interpretation of the results in terms of arousal-mood theory. Future research examining the effects of music on cognitive performance should collect behavioral and physiological measures of arousal

alongside mood to further evaluate this theory. Fourth, the time-scale of this experiment was relatively short compared to that taken to complete many real-world work tasks. It is possible that the music tested here may have different effects at longer time scales. Finally, despite placing consistent and measurable demands on cognition, the flanker task is relatively simplistic. Indeed, most participants made few errors across the Flanker conditions (mean correct answers ± SD for all participants, regardless of audio condition: NoGo, 15.12 ± 4.93; Congruent, 17.66 ± 1.08; Incongruent, 16.65 ± 3.13; Neutral, 17.48 ± 1.63). This ceiling effect might account for the lack of significant effects of music background condition on Flanker accuracy. Future studies should examine music's cognitive effects with a greater variety of tasks and/or more naturalistic work settings (e.g., real world problem solving, creative writing or design, critical analysis, etc.) to more precisely define how and when listening to music can be beneficial.

In conclusion, here we show that instrumental music intentionally composed to support attentional focus and concentration during work—comprising strong rhythm, simple tonality, broad spectral energy, and moderate dynamism—improves mood and increases processing speed during a cognitively demanding task that requires selective attention. These effects were demonstrated in comparison to more minimalistic music similarly advertised to improve attentional focus, popular music, and typical background noise in a calm office environment. This work has real-world implications for providing the general population with effective and affordable strategies to regulate mood and performance during routine work tasks often experienced as emotionally and physically taxing.

## Supporting information

**S1 File. Supplementary text that includes (i) a brief description of each of the 23 musical features used to characterize the music used in this work along with the MIRtoolbox / Matlab function calls used to extract them; (ii) S2-S4 Tables; and (iii) S1-S5 Figs.**
(PDF)

**S1 Table. Table with the track names and artists, set arrangements, and musical features for each track in each audio condition.**
(XLSX)

**S1 Data. All behavioral data analyzed using JASP.**
(XLSX)

**S2 Data. All behavioral data analyzed using R.**
(XLSX)

**S1 Audio. Track used for the work flow condition.**
(M4A)

**S2 Audio. Track used for the work flow condition.**
(M4A)

**S3 Audio. Track used for the work flow condition.**
(M4A)

**S4 Audio. Track used for the work flow condition.**
(M4A)

**S5 Audio. Track used for the office noise condition.**
(M4A)

## Acknowledgments

We thank David Poeppel for early feedback on this manuscript.

## Author Contributions

**Conceptualization:** Joan Orpella, Daniel Liu Bowling, Concetta Tomaino, Pablo Ripollés.

**Data curation:** Joan Orpella, Pablo Ripollés.

**Formal analysis:** Joan Orpella, Pablo Ripollés.

**Funding acquisition:** Pablo Ripollés.

**Investigation:** Joan Orpella, Daniel Liu Bowling, Pablo Ripollés.

**Methodology:** Joan Orpella, Daniel Liu Bowling, Concetta Tomaino, Pablo Ripollés.

**Project administration:** Joan Orpella, Pablo Ripollés.

**Resources:** Pablo Ripollés.

**Supervision:** Joan Orpella, Daniel Liu Bowling, Concetta Tomaino, Pablo Ripollés.

**Validation:** Daniel Liu Bowling, Pablo Ripollés.

**Visualization:** Joan Orpella, Pablo Ripollés.

**Writing – original draft:** Joan Orpella, Daniel Liu Bowling, Pablo Ripollés.

**Writing – review & editing:** Joan Orpella, Daniel Liu Bowling, Concetta Tomaino, Pablo Ripollés.

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
