## [Decision Letter · Decision Letter 0]

26 Aug 2024

PONE-D-24-26875Effects of Music Advertised to Support Focus on Mood and Processing SpeedPLOS ONE

Dear Dr. Ripolles,

Thank you for submitting your manuscript to PLOS ONE. After careful consideration, we feel that it has merit but does not fully meet PLOS ONE’s publication criteria as it currently stands. Therefore, we invite you to submit a revised version of the manuscript that addresses the points raised during the review process.

Two experts in the field have carefully reviewed the manuscript entitled "Effects of Music Advertised to Support Focus on Mood and Processing Speed" . Both reviewers have made observations that need to be addressed (see below).

In light of these reviews, I am requesting a minor revision and resubmission, in which you will need to respond to each point made in the reviews.

We look forward to receiving your revised manuscript.

Kind regards,

Bruno Alejandro Mesz, Ph.D.

Academic Editor

PLOS ONE

Journal Requirements:

"This work was funded by a grant from the company that provided the tracks for the work flow condition to Dr. Ripolles."

"This work was funded by a grant from the company that provided the tracks for the work flow condition to Dr. Ripolles."

"Dr. Tomaino serves as the music therapy advisor at the company that provided the tracks for the work flow condition. Dr. Bowling serves as the neuroscience advisor at the company that provided the tracks for the work flow condition. This work was funded by a grant from the company that provided the tracks for the work flow condition to Dr. Ripolles. Dr. Orpella has no competing interests."

Reviewers' comments:

Reviewer's Responses to Questions

**Comments to the Author**

1. Is the manuscript technically sound, and do the data support the conclusions?

Reviewer #1: Yes

Reviewer #2: Yes

2. Has the statistical analysis been performed appropriately and rigorously? 

Reviewer #1: Yes

Reviewer #2: Yes

3. Have the authors made all data underlying the findings in their manuscript fully available?

Reviewer #1: Yes

Reviewer #2: Yes

4. Is the manuscript presented in an intelligible fashion and written in standard English?

Reviewer #1: Yes

Reviewer #2: Yes

5. Review Comments to the Author

Reviewer #1: ### Summary of paper

The current paper evaluates the effects of music advertised to support attentional focus on mood and performance of participants on a cognitively demanding task (the flanker test). The study stands out from previous work as it focuses on evaluating the effect of music in task performance. It is also one of the few studies where the effect is regarding music that is being listened to while the task is being performed (instead of before). The study also investigates the effect of the music in mood and the liking of it by the participants, in order to evaluate possible mechanisms through which changes in task performance may be explained (in particular, the arousal-mood theory). In a between-subjects design, they compared two music sets advertised to improve performance (”work flow” and “deep focus”) with two control conditions: popular music and office background noise. Groups were assessed to be comparable in terms of basic demographic, music-related, and mental health variables. Results showed that only the “work flow” music yielded performance improvement as decreased RT. Music condition had no effect on task accuracy. The paper clearly states the set of a priori statistic analyses by introducing them in the methods section and then exploratory analyses introduced during the result section. A priori analyses were: similarity of participant groups, effect of audio condition on mood change, effect of audio condition on performance accuracy, and effect of audio condition on performance speed considering sensitivity to musical reward, music training and basal psychological distress. Results showed a main effect of anxiety and sensitivity to musical reward for RT as well as decreased RT for the “Work flow” condition. Exploratory analyses focused on looking for possible modulators of the performance improvement effect. Tests were performed for the effect of basal levels of depression, anxiety and stress as well as mood change. Basal levels of depression, anxiety and stress did not show an effect on performance improvement, but mood change did. Greater mood change correlated with lower RTs over time. Additionally, they verified that changes in mood were not predicted by an interaction of music condition and music familiarity or pleasure. Yet, changes in mood were related to musical pleasure. These results are used to hypothesize that the speed improvement in the task may be due to the music’s ability to upregulate arousal and positive affect simultaneously, which seemed to happen mainly with “work flow” music, as it yielded high pleasure as well as high groove ratings, contrasting other music conditions.

The conclusions seem to be well supported by the results and the analysis. The three supplementary tables contain all the raw data mentioned in the paper (acoustic features of the stimuli, background data of the participants and individual trial responses).

### Main comments

Here I detail main concerns regarding presentation.

1. One of the main concerns is that I was unable to find the supplementary figures (Figures S1-S4).

2. The explanation on how the stimuli was selected lacks detail. In the Stimuli section (l117 - l161) it is said that the tracks were sampled from a larger set. No detail is provided on the sampling method. Later on, in the discussion (l489) it is said “While these sets were assembled with the intention of maintaining uniformity in musical features”. This should be explained in the stimuli section. Moreover, some further analysis on how similar the music sets were (e.g.: Silhouette score) could better illustrate the picture.

3. In the results for Group categorization (l324), it is stated: "Music genre preferences varied within groups but were comparable across them”. Here, it is not clear what the music genre preferences refer to or how they are comparable. The text references Table S2, which contains the Pleasure, Familiarity and Groove ratings. If this is what the text refers too, then this is explained in the discussion section, when Bayesian stats are presented (l513-l517). These results should be presented in the results section.

4. In the discussion, a main difference is stated between the “Work flow” condition and other conditions based on musical attributes of the music; namely, “musical features closely associated with perceived arousal, such as spectral flux” as well as strong rhythmic features such as pulse clarity. Another important difference is given by the “Work flow” sets being comprised of a single coherent track, while the other musical conditions (excluding “Office”) having multiple tracks with silence in between. I would argue that the entrainment allowed by a coherent continuous track could be part of the reasons for improved RT, or the cognitive load of adapting to a changing auditory environment a hindrance to the benefits of music.

### Minor comments

- p6, 155: "The office noise track was generated with the following 155

settings on the sound generator web page" - I would add an explanation on the criterion by which these settings were selected.

- p7, 164: I would report MATLAB version used, as MIRToolbox has shown to work differently with different MATLAB versions.

- p11, 267: "The same procedure was used..." - I am confused on which procedure this is as this

is for only post-measured variables instead of pre-post (which is what the last analysis referred to). Is it the same analysis as the previous paragraph?

- p17: "it also included the three-way interaction between audio condition,

trial number, and DASS21 anxiety subscore." - Does this model also contain

the previous terms with these variables (i.e.: AudioCondition*TrialNumber +

DASS21)?

- p20, 469: DB and CT introduced out of nowhere

Reviewer #2: General comments

In this study, the authors wanted to test if music advertised as able to improve work flow or to engage listeners in deep focus affects a cognitively demanding task (flanker test). They also tested if this music affected the listeners’ mood and state.

They test this by running an online experiment on MTurk (n~200). The experiment consisted of a series of music, mental health, and mood-related questionnaires followed by a flanker task. This flanker task had to be solved while listening to three types of background music (work flow, deep focus, and pop hits) and a control condition (calm office noise).

They found that the Work Flow music significantly improved mood (as measured by PANAS) and differentially decreased RTs with time on the flanker task. They did not find any effect on participants’ accuracy on the flanker task.

The manuscript is clear in presenting the research question and all the relevant previous work in the area. The stimuli selection and the experimental procedure and clearly and thoroughly described in the manuscript. The experimental design and the data collected were consistent with the hypothesis they wanted to test. The statistical tools used were generally correct and well-interpreted by the authors.

However, there are several points the authors should address for this manuscript to be ready for publication. The main concern is that the significant triple interaction (shown in Figure 6) seems to show a more negative slope of the RTs for the work flow condition but the predicted RTs are larger for most of the trials (~60 out of 72).

The authors properly disclosed their conflict of interest.

Particular comments

P4-L85: Remove the “very” in “very different neural and behavioral responses” unless there is a quantification of this expectation.

P4-L94: I will not consider N=196 as a large-scale experiment. Please remove this adjective or add references that support this claim.

P5-L110: Why would the authors want to be able to detect a medium effect size with the given statistical power? Please provide a rationale for this decision. What was the dependent variable in your power analysis? The manuscript is full of statistical tests and it should be clear for which one they planned the power.

P5-L110: How did the authors estimate the variability for the power analysis? Did they use data from a pilot study or an estimation based on previous studies? Please provide additional information on this matter to the manuscript.

P5-L131: Add a reference to support this affirmation: “Given prior research on the effects of listening to pop music during work, we expected this condition to negatively impact on-task performance.“

P7-L168: Supplementary material with details on how the musical features were extracted was not provided in the manuscript. Please add this document to the revised version.

P11-L271: ANOVAs assume that residuals are normally distributed, something that rarely happens on count data, especially if you experience any floor or ceiling effect. In the particular case of the data of this study, this assumption is not met mainly because the number of correct responses is not only a count variable but it is bounded between 0 and 18 with most of its mass (for most subjects) close to 18. The authors could explore models for bounded count data (generalized linear mixed-effect models for Poisson or Negative Binomial) to properly model the participant’s responses.

A more direct approach could be to replicate the model structure of the RTs but using correct/incorrect as the dependent variable and “logit” as the link function of a generalized linear mixed effect model (although this seems to have convergence issues with your data). This could be the reason for the non-significant results on page 16 (L388 to L397).

P11-L276: I do not think the substantial variability is a reason to use mixed-effect models. If you think that is the case please provide a quantifiable definition of substantial and references that justify the relationship with that and the recommendation to use linear mixed models.

P13-L314: There is an extra “)” after “noise”.

P13-L319: Figure S1 was not in the manuscript. Please add this document to the revised version.

P13-L320: A BF of 2.11 is usually not considered strong evidence for a given hypothesis. Please better explain the group differences in musical training.

P13-L324: Table S2 was not in the manuscript. Please add this document to the revised version.

P13-L327: Add the individual data points to Figure 3A.

P13-L327: It is not clear if the only statistically significant difference condition was work flow. I assume that when it is not mentioned the results were nonsignificant but please, if that is the case, make it explicit.

P13-L330: Clarify how you corrected for multiple comparisons when running the one-sample t-tests.

P14-L355: Add the individual data points to Figure 4.

P14-L361: When using null-hypothesis statistical testing the results are dichotomical: Differences are either different from zero (given a significance level set by the selection of alpha) or not. The authors should remove the “marginally less” statement from the text and the use of * for p=0.88 from the figure. This last issue is especially problematic since it could be mistaken by the more common use of * for p<0.05.

P14-L369: How did the authors phrase the question about familiarity with the “office noise” condition?

P16-L384: Figure S5 was not in the manuscript. Please add this document to the revised version.

P16-L388: Figure S2 was not in the manuscript. Please add this document to the revised version.

P17-L414: Figure S4 was not in the manuscript. Please add this document to the revised version.

P17-L418: Authors should provide a table with all the estimated parameters of the fitted model (with CIs). I would recommend they use the modelsummary R package.

P17-L421: Please provide a p-value after “(slope; β=-0.421)”.

P17-L423: The use of the wording “task performance” could be misleading since what the authors observed is a differential effect of the time on the response time but not on the accuracy of the responses. Please rephrase the last sentence of that paragraph.

P18-L425: A non-significant result could mean that there is no difference only if the sample size was determined with an apriori power analysis, otherwise this could attributed to a lack of statistical power. The non-significant three-way interaction does not “indicate” that the observed effect of work flow music on task performance over time was independent of anxiety severity as assessed by the DASS-21”

P18-L432: Authors should add a title to the color bar on Figure 6.

P18-L432: Two things in Figure 6 are interesting and not addressed in the manuscript: 1- Although larger PANAS scores are associated with a faster improvement of the RTs, it also seems to make the participants respond slower in the work flow condition for the first trials. 2- The predicted RTs cross around trial number 60, and trial number 72 the difference in RTs is quite small compared to the ones at the beginning of the experiment. The authors should explain both these effects in more detail.

P18-L447: If I am not mistaken, the authors could not affirm that “These mood effects were not explained by differences between audio conditions in self-reported musical pleasure” since there is a main effect of pleasure on PANAS change. I assume the previous statements refer to the interaction being non-significant, but given that the mean pleasure level is not the same for all the musical stimuli, music's effect on mood could, in fact, be partially mediated by pleasure.

6. PLOS authors have the option to publish the peer review history of their article (what does this mean?). If published, this will include your full peer review and any attached files.

Reviewer #1: **Yes: **Martin Alejandro Miguel

Reviewer #2: **Yes: **Ignacio Spiousas

---

## [Author Response · Author response to Decision Letter 0]

13 Oct 2024

We thank the editor and reviewers for their comments. We have provided answers to all of them

---

## [Decision Letter · Decision Letter 1]

28 Nov 2024

PONE-D-24-26875R1Effects of Music Advertised to Support Focus on Mood and Processing SpeedPLOS ONE

Dear Dr. Ripolles,

Thank you for submitting your manuscript to PLOS ONE. After careful consideration, we feel that it has merit but does not fully meet PLOS ONE’s publication criteria as it currently stands. Therefore, we invite you to submit a revised version of the manuscript that addresses the points raised during the review process.

Both reviewers have now accepted your manuscript. However, one of them believes that there are still two points that require further clarification. After you consider these points, i will submit my decision without sending it again for further revision.

We look forward to receiving your revised manuscript.

Kind regards,

Bruno Alejandro Mesz, Ph.D.

Academic Editor

PLOS ONE

Journal Requirements:

Reviewers' comments:

Reviewer's Responses to Questions

**Comments to the Author**

1. If the authors have adequately addressed your comments raised in a previous round of review and you feel that this manuscript is now acceptable for publication, you may indicate that here to bypass the “Comments to the Author” section, enter your conflict of interest statement in the “Confidential to Editor” section, and submit your "Accept" recommendation.

Reviewer #1: All comments have been addressed

Reviewer #2: All comments have been addressed

2. Is the manuscript technically sound, and do the data support the conclusions?

Reviewer #1: Yes

Reviewer #2: Yes

3. Has the statistical analysis been performed appropriately and rigorously? 

Reviewer #1: Yes

Reviewer #2: Yes

4. Have the authors made all data underlying the findings in their manuscript fully available?

Reviewer #1: Yes

Reviewer #2: Yes

5. Is the manuscript presented in an intelligible fashion and written in standard English?

Reviewer #1: Yes

Reviewer #2: Yes

6. Review Comments to the Author

Reviewer #1: All comments have been properly addressed. The missing supplementary material is sound, complementing the main text. The clustering analysis performed on the selected music is satisfactory to understand that the sets in the "deep focus" playlist are similar with each other. This is similar for the sets in the "work flow" playlist, with the exception of one track, which is still better clustered with the "work flow" set.

Reviewer #2: The authors have thoroughly addressed all comments, concerns, and suggestions. They accepted the majority and, in cases where they did not, provided a sufficiently robust rationale.

However, I believe there are still two points that require further clarification:

1- The rationale for fitting mixed-effects models remains unclear. Random effects should only be included when residuals are correlated, typically due to the way units are sampled (i.e., randomly). I recommend that the authors either justify the use of mixed-effects models appropriately or refrain from providing a justification if it cannot be done accurately.

2- The model and analysis of the triple interaction (Figure 6) should be included as supplementary material, with a reference to it added in the main text.

7. PLOS authors have the option to publish the peer review history of their article (what does this mean?). If published, this will include your full peer review and any attached files.

Reviewer #1: **Yes: **Martin Alejandro Miguel

Reviewer #2: **Yes: **Ignacio Spiousas

---

## [Author Response · Author response to Decision Letter 1]

1 Dec 2024

We have addressed the final issues raised by reviewer 2

---

## [Editor Report · Decision Letter 2]

5 Dec 2024

Effects of Music Advertised to Support Focus on Mood and Processing Speed

PONE-D-24-26875R2

Dear Dr. Ripolles,

We’re pleased to inform you that your manuscript has been judged scientifically suitable for publication and will be formally accepted for publication once it meets all outstanding technical requirements.

Kind regards,

Bruno Alejandro Mesz, Ph.D.

Academic Editor

PLOS ONE
---

## [Editor Report · Acceptance letter]

10 Jan 2025

PONE-D-24-26875R2 

PLOS ONE

Dear Dr. Ripolles, 

I'm pleased to inform you that your manuscript has been deemed suitable for publication in PLOS ONE. Congratulations! Your manuscript is now being handed over to our production team.

Kind regards, 

on behalf of

Dr. Bruno Alejandro Mesz 

Academic Editor

PLOS ONE